# Temporal Relationship-Aware Treadmill Exercise Test Analysis Network for Coronary Artery Disease Diagnosis

**DOI:** 10.3390/s24092705

**Published:** 2024-04-24

**Authors:** Jianze Wei, Bocheng Pan, Yu Gan, Xuedi Li, Deping Liu, Botao Sang, Xingyu Gao

**Affiliations:** 1Institute of Microelectronics, Chinese Academy of Sciences, Beijing 100029, China; weijianze@ime.ac.cn (J.W.); panbocheng@ime.ac.cn (B.P.); 2Cardiology Department, Beijing Hospital, Beijing 100730, China; ganyu4016@bjhmoh.cn (Y.G.); lixuedi7792@bjhmoh.cn (X.L.); sangbotao23@mails.ucas.ac.cn (B.S.); 3National Center of Gerontology, National Health Commission Institute of Geriatric Medicine, Chinese Academy of Medical Sciences, Beijing 100730, China; 4University of Chinese Academy of Sciences, Beijing 100006, China

**Keywords:** machine CAD diagnosis, contextual learning, deep learning

## Abstract

The treadmill exercise test (TET) serves as a non-invasive method for the diagnosis of coronary artery disease (CAD). Despite its widespread use, TET reports are susceptible to external influences, heightening the risk of misdiagnosis and underdiagnosis. In this paper, we propose a novel automatic CAD diagnosis approach. The proposed approach introduces a customized preprocessing method to obtain clear electrocardiograms (ECGs) from individual TET reports. Additionally, it presents TETDiaNet, a novel neural network designed to explore the temporal relationships within TET ECGs. Central to TETDiaNet is the TETDia block, which mimics clinicians’ diagnostic processes to extract essential diagnostic information. This block encompasses an intra-state contextual learning module and an inter-state contextual learning module, modeling the temporal relationships within a single state and between states, respectively. These two modules help the TETDia block to capture effective diagnosis information by exploring the temporal relationships within TET ECGs. Furthermore, we establish a new TET dataset named TET4CAD for CAD diagnosis. It contains simplified TET reports for 192 CAD patients and 224 non-CAD patients, and each patient undergoes coronary angiography for labeling. Experimental results on TET4CAD underscore the superior performance of the proposed approach, highlighting the discriminative value of the temporal relationships within TET ECGs for CAD diagnosis.

## 1. Introduction

Coronary artery disease (CAD) is a cardiovascular condition characterized by atherosclerosis, which restricts the blood flow to the coronary arteries, thereby posing a severe threat to an individual’s life [1,2]. In 2019, the World Health Organization (WHO) listed CAD as one of the top ten causes of death (https://www.who.int/news-room/fact-sheets/detail/the-top-10-causes-of-death, accessed on 6 February 2024), which draws widespread attention around the world. Coronary angiography (CAG) is the gold standard for the diagnosis of CAD [3,4,5]. Nevertheless, as an invasive CT test, CAG has risks such as blood vessel injury, excessive bleeding, heart attacks, etc. [4]. This forced researchers to develop the treadmill exercise test (TET) [6,7], which is a non-invasive procedure without the need for catheter insertion or exposure to contrast agents. However, the TET results are susceptible to various influencing factors, including physical fitness, age, and gender. Additionally, the absence of a quantitative absolute standard for CAD diagnosis, compounded by individual patient differences, presents a significant challenge for clinicians attempting to interpret TET results [8]. Consequently, clinicians must invest considerable time in manually analyzing the TET results to mitigate the risk of misdiagnosis or underdiagnosis.

In recent years, the rapid development of artificial intelligence has yielded impressive milestones in a variety of fields. Some research has attempted to employ machine learning methods for CAD diagnosis [9,10,11]. Lee et al. [12] learn a new feature representation by integrating 30 pieces of information about the patient’s age, gender, and maximum metabolic equivalents and employ multiple machine learning algorithms as classifiers for diagnosis. Yilmaz et al. [13] reconstruct a series of numerical amplitudes from the electrocardiogram (ECG) and segment them into P, QRS, and T waves for subsequent feature extraction and diagnosis prediction. Lu et al. [14] employ a generative model for data augmentation to address the class imbalance problem and develop a neural oblivious decision ensemble method to predict whether the patient has CAD. These methods automatically diagnose CAD based on ECG images or ECG numerical vectors captured in a single state for CAD diagnosis, demonstrating the potential of machine learning in disease diagnosis. However, the clinical diagnostic experience based on TET suggests that dynamic variations in ECGs during different states are a pivotal clue for CAD diagnosis.

To prompt the development of machine CAD diagnosis, we built a new TET dataset named TET4CAD by collecting TET ECG images from 416 patients and labeling them according to their CAG results. Specifically, we invited 416 patients, i.e., 192 CAD patients, and 224 non-CAD patients for data collection and asked them to undergo both TET and CAG for free. Patients underwent TET using the standard Bruce protocol, and their TET results were reported using CASE produced by GE Medical Systems Information Technologies Inc. (Chicago, IL, USA). For the TET result of each patient, we sampled one 12-lead ECG image for each state (pretest, exercise, and recovery) and integrated these three ECG images as the TET data of the patient. Meanwhile, CAG was performed for each patient using the Allura Xper FD20 produced by Philips, Amsterdam, The Netherlands. A specialist clinician was invited to diagnose the patients based on their CAG results, and the diagnostic conclusions served as the ground-truth labels for the presented dataset. In addition, we also recorded the age and gender of each patient to facilitate subsequent experiments. In summary, this dataset records 12-lead ECG images of 416 patient under pretest, exercise, and recovery states by following TET and annotates their diagnostics based on CAG. Based on the above, this dataset preserves the dynamics of the ECG in different states for better machine CAD diagnosis. Here, we take this dataset as the benchmark dataset to validate the effectiveness of the proposed method.

Based on the TET4CAD dataset, we propose an automatic CAD diagnostic approach to investigate the dynamic variations in the 12-lead ECG of TET by mimicking the clinical process of diagnosing CAD. The proposed approach applies an automatic end-to-end pipeline of “preprocessing–feature extraction–classification”. In the preprocessing step, the proposed approach initially converts the PDF file with three 12-lead ECGs into a group of images, and each image has a resolution of “1080×640”. Subsequently, we strategically crop the converted images according to the preset coordinates to filter out the content irrelevant to ECGs. Finally, a series of image processing techniques are utilized to remove redundant backgrounds such as grid lines. The remaining 12-lead ECG images are saved as the preprocessed results. In the feature extraction step, we develop a novel TET-based diagnosis network termed TETDiaNet to explore the dynamic relationships within TET ECGs. The core of the network is the TETDia block, which is able to explore the contextual relationships within the same state and between different states for each lead. To achieve this, we split the 12-lead ECGs into 12 patches according to the lead locations and assign them to different channels. TETDia block leverages an intra-state contextual learning module to investigate the temporal relationships within individual ECGs under a single state and an inter-state contextual learning module to learn the correlations between different states. TETDiaNet captures these relationships like a clinician to learn discriminative features for CAD diagnostics. In the classification step, a three-layer multilayer perceptron is employed for disease prediction based on the features learned by TETDiaNet. The proposed approach works like a clinician to diagnose disease according to TET by exploring the temporal relationships within individual ECGs under a single state and the correlations within individual ECGs between different states. The related experimental results demonstrate the effectiveness of the proposed approach.

The main contributions are summarized as follows.

The paper presents a new dataset, named TET4CAD, consisting of 192 CAD samples and 224 non-CAD samples. This dataset imitates TET and retains multiple 12-lead ECGs in various states to support the development of machine CAD diagnosis.We present a preprocessing module for the newly built TET4CAD dataset. It highlights important ECGs in TET results with low information density by removing the redundant backgrounds irrelevant to CAD diagnosis.We propose a new machine CAD diagnostic model named TETDiaNet to automatically predict whether a patient has CAD according to his/her TET results. TETDiaNet learns discriminative features from the TET results by exploring the temporal relationships within individual ECGs under a single state and the correlations within individual ECGs between different states.The superior performance of the proposed method on the self-built dataset illustrates that our proposed approach is able to accurately predict diagnostic results according to TET, effectively reducing misdiagnosis and underdiagnosis.

The subsequent sections of this paper are organized as follows. Section 2 briefly revisits machine learning methods for CAD diagnostics. Section 3 and Section 4 explicate the details of the newly built TET4CAD dataset and technical intricacies of the proposed method, respectively. Section 5 presents the experimental outcomes on the self-built dataset along with the empirical analysis. The paper concludes with Section 6.

## 2. Related Work

Coronary artery disease (CAD) is characterized by the accumulation of atherosclerotic plaques in the walls of the coronary artery “tree”. While this can result in restricted blood flow to the heart muscle (myocardium) due to the significant narrowing (stenosis) of the coronary artery lumen [15], even mildly stenotic plaques pose a significant risk to the affected patient [16].

To help doctors to improve the diagnostic efficiency, researchers have attempted to use machine learning methods for CAD diagnosis. The existing methods for CAD diagnosis can be divided into three categories: traditional machine learning, ensemble learning, and deep learning methods. Earlier research mainly adopted traditional machine learning, relying on manual feature engineering and statistical learning principles for pattern recognition. These methods have limitations regarding the data scale and feature selection. Intermediate research employed ensemble learning methods, constructing more powerful classifiers by ensembling multiple base learners. In recent years, influenced by the rise of deep learning [17,18], deep learning models have been widely applied to ECG analysis, enabling end-to-end training and significantly improving diagnosis through hierarchical feature learning. With the growth of datasets and computing power, deep learning is regarded as the most promising technology for CAD diagnosis.

### 2.1. Traditional Machine Learning Methods

Traditional machine learning methods rely on manual feature extraction from ECGs and use statistical learning principles for CAD pattern recognition [19,20]. Common algorithms include support vector machine (SVM), logistic regression, decision trees, etc. These methods can automatically learn correlations between ECG and CAD to categorize conditions. Regarding traditional machine learning methods, Beunza et al. [21] used demographic, lifestyle, and laboratory features with SVM to predict CAD, achieving an AUC of 0.75. Gola et al. [22] used six imputed data features with an SVM model to make predictions, obtaining an AUC of 0.92. Several successful automated and semi-automated algorithms have been developed to extract the artery centerlines, analyze the vessels, and detect visual clues related to pathological changes related to atherosclerosis. Some of these algorithms were designed specifically to detect calcified plaques [23] or plaques in general [24]. Although these algorithms report high scores for abnormality detection, they are generally rule-based or based on conventional machine learning, which is heavily reliant on hand-crafted tuning for model parameters.

### 2.2. Ensemble Learning Methods

Ensemble learning methods such as bagging, boosting, and stacking combine multiple base learners and aggregate them to obtain a more powerful ensemble classifier for improved ECG analysis [25,26,27]. Baskaran et al. [28] used demographic, risk factor, and medication history features with XGBoost [29], achieving an AUC of 0.705 and sensitivity of 89.2%. Al’Aref et al. [30] utilized demographic, disease, and physical indicator features, with algorithms like AdaBoost [31], XGBoost, random forest [32], and logistic regression, resulting in an AUC of 0.927. Du et al. [33] used demographic, medical history, and procedure features with an ensemble XGBoost, obtaining an AUC of 0.94. Ensemble learning methods improve traditional machine learning’s classification performance but still require manual feature selection rather than end-to-end feature learning and classification.

### 2.3. Deep Learning Methods

In recent years, deep learning methods, especially convolutional neural networks (CNNs), have made great strides in ECG analysis and CAD diagnosis [34,35,36]. Deep neural networks are inspired by the brain’s structure and achieve disease diagnosis and classification through hierarchical feature learning. Typically, the effectiveness of a deep learning system hinges on the presence of substantial volumes of well-annotated training data specific to the model in question. Through streamlined data curation [37] and meticulous annotation processes [38], the integration of a deep learning system into a clinical imaging workflow becomes a seamless endeavor. For instance, multilayer perceptrons optimize the connection weights via gradient descent and excel in discovering nonlinear patterns in healthcare data [39].

For large-scale datasets, simple neural networks are insufficient and deep networks like CNNs are needed. CNNs automatically learn feature representations through convolution layers without manual feature engineering. Studies show the increasing prevalence of various deep networks for CAD diagnosis. Deep learning models can directly extract diagnostic features from raw ECG data, enabling end-to-end learning in a purely data-driven manner. For example, multilayer perceptron models achieved almost 100% classification accuracy on the Z-Alizadeh Sani dataset [40]. In summary, deep learning methods overcome the limitations of traditional machine learning by using hierarchical feature learning to precisely identify CAD from ECGs, representing the current research hotspot and development trend in this field.

Different from the above methods, this paper presents a new dataset (TET4CAD) containing the dynamic variation in the ECG under different states. Correspondingly, the proposed TETDiaNet perceives this variation by investigating the temporal relationships within individual ECGs under a single state and the correlations within individual ECGs between different states for CAD diagnosis.

## 3. TET4CAD Dataset

To our knowledge, the majority of clinicians rely on treadmill exercise test (TET) results for coronary artery disease (CAD) diagnosis, scrutinizing the dynamic variations in electrocardiograms (ECGs) across different states. Despite the significance of this a priori clinical experience, existing methods do not follow this priori experience since there is no publicly available dataset providing ECGs recorded under diverse physiological states This data absence hinders the performance improvement of machine CAD diagnosis. In a proactive effort to advance machine CAD diagnosis, we collected the TET4CAD dataset, and this work was approved by Beijing Hospital (protocol number: 2024BJYYEC-KY043-01).

### 3.1. Data Collection

The dataset comprised information gathered from 614 patients presenting with heart complaints at the Beijing Hospital. A skilled clinician administered both a treadmill exercise test (TET) and a coronary angiography (CAG) to determine the presence of coronary artery disease (CAD).

Patients underwent the TET following the established Bruce protocol, in alignment with the current guidelines for those under suspicion of CAD, as shown in Figure 1. All study participants had a Duke treadmill risk score of <−10. Retrospectively, TET reports were obtained from CASE, produced by GE Medical Systems Information Technologies Inc., for patients who underwent both TET and CAG within the same month. In addition, reports with non-periodic, noise-containing electrocardiogram (ECG) signals resulting from TET disturbances were excluded from the study with confirmation from cardiologists, resulting in a final dataset of 416 TET reports. Considering that clinicians make a diagnosis based on ECGs in different movement states, the TET4CAD dataset simplifies the TET report by sampling one 12-lead ECG for each of the pretest, exercise, and recovery states, respectively. Thus, each TET datum contains three ECGs under different states for CAD diagnosis. Table 1 outlines the baseline characteristics of the 416 patients included in the TET4CAD dataset.

### 3.2. Data Annotation

To promote the application of machine learning in CAD diagnostics, we have meticulously provided ground-truth labels for the TET4CAD dataset.

The ground-truth labeling process for TET reports was meticulously established based on CAG reports. Expert cardiologists assumed the responsibility of meticulously assessing the extent of lumen diameter narrowing in specific coronary arteries, including the right coronary artery, left main coronary artery, left anterior descending artery, and left circumflex artery. Reports that revealed an obstruction of 50% or more in any of these arteries were considered indicative of patients with obstructive coronary artery disease (CAD). Conversely, cases where the detected obstructions fell below the 50% threshold were categorized as non-obstructive CAD cases.

Following this process, in the final dataset of 416 TET reports, 192 patients were identified as having obstructive CAD (here, we assign CAD labels to them), while 224 patients had non-obstructive CAD (non-CAD labels are assigned to them).

### 3.3. Dataset Analysis

Compared to the datasets utilized in prior studies [12,13,14], the TET4CAD dataset uniquely encompasses the dynamic variations observed in ECGs across different states. This comes from our observations of clinicians diagnosing CAD. Specifically, patients manifest distinct ECG patterns throughout various states of TET. This dynamic change serves as a crucial diagnostic clue, routinely taken into consideration by clinicians when evaluating CAD based on TET reports. The TET4CAD dataset meticulously preserves this dynamic nature by retaining a set of 12-lead ECGs for each distinct state, providing a comprehensive representation of the evolving cardiac response during TET.

Furthermore, the distribution of different groups within our dataset is consistent with the demographics of CAD patients observed in Beijing Hospital. Table 1 reports the gender and age distributions of the proposed TET4CAD dataset. To our knowledge, the age and gender distributions are roughly similar to those of the datasets used in [12,13,14]. Specifically, there are 299 male and 117 female patients in TET4CAD according to Table 1, which maintains a general balance between the different genders. In terms of the age distribution, it shows a Gaussian distribution, and patients in the age ranges of (50,60] and (60,70] dominate the TET4CAD dataset. Based on our experience with the clinical diagnosis of CAD, these patients represent the main groups in CAD diagnosis and are most susceptible to CAD. Figure 2 plots the age distribution of the TET4CAD dataset, which is consistent with the age distribution of patients attending the Beijing Hospital. For the proposed TET4CAD dataset, there exist potential biases regarding age and gender, which could limit the generalizability of the diagnosis model trained on it. In the future, we will explore (1) the relationship between CAD and gender, as well as age, and (2) a diagnosis model with better generalization capabilities based on the proposed TET4CAD dataset.

## 4. Method

In this paper, we present an automatic CAD diagnostic method to explore the temporal relationships of TET. ECGs derived from TET reports are a novel type of ECG that can record the electrical signals of the heart in different states, including pretest, exercise, and recovery. A TET ECG can reflect the functional and metabolic changes of the heart, thus providing more information for CAD diagnosis.

In the newly built TET4CAD dataset, each sample provides 12-lead ECGs under three states for CAD diagnosis. However, these recorded ECGs are in PDF format, containing redundant elements like grids in the background, making them challenging to process. Therefore, the proposed method initiates with essential preprocessing steps. The preprocessing involves converting the PDF files into an image format, generating a group of images with a resolution of “1080×640”. Subsequently, the images undergo cropping to retain only the ECG region of interest and enhancement to mitigate the impact of background elements such as grids. More details of the preprocessing steps are provided in Section 4.1.

For a more rational interpretation of the ECGs of individual leads, the proposed method introduces TETDiaNet to investigate the temporal relationships of TET. TETDiaNet first divided the 12-lead ECGs into 12 patches according to the lead position and assigned them to different channels. Each channel corresponds to a single-lead ECG under a single state. Considering that clinicians diagnose CAD based on the dynamic changes in ECGs between different states, we design the TETDia block as the basic unit of TETDiaNet to explore the temporal relationships within individual ECGs under a single state and the correlations within individual ECGs between different states. Section 4.2 describes the technical details of TETDiaNet. In addition, Section 4.2.3 and Section 4.3 detail the structure of the classifier and the object function for the proposed method.

In summary, the proposed method consisting of a preprocessing module, a TETDiaNet, and a classifier captures these relationships like a clinical doctor, to learn the discriminative features for CAD diagnosis.

### 4.1. Preprocessing

The preprocessing module converts each simplified TET report in PDF format into a set of images. It consists of image conversion, image cropping, and image enhancement, as shown in Figure 3.

#### 4.1.1. Format Conversion and Image Cropping

To correctly identify the state of each ECG, optical character recognition (OCR) techniques are utilized to scan the patients’ ECG PDF documents page by page, recognizing all textual information contained on each page. Then, three keywords are utilized to find the state of each page, namely pretest, exercise, and recovery. Specifically, a matching algorithm searches the OCR results against predefined keywords, acquiring the page numbers for the three state pages. Of note, noise filtering was applied to the OCR results to improve the matching accuracy by removing non-text noise.

For each PDF file with defined states, we converted it into a group of three images in PNG format. Each image corresponded to a 12-lead ECG for one state and had a resolution of “1080×640”. Then, the preprocessing module crops the images using predefined image coordinates to remove the non-ECG regions from the converted image. Using format conversion and image cropping, each sample in the database is transformed into a group of three images with different states.

#### 4.1.2. Image Enhancement

TET pages contain rich information for CAD diagnosis. Some of the information, such as grid lines, is helpful for clinical diagnosis but can hinder machine diagnosis. To solve this problem, the preprocessing module leverages a series of image processing methods to enhance the image.

Specifically, as shown in Figure 3, the original ECG image is first converted into the grayscale space. This behavior simplifies the image so that it contains only grayscale information, providing a basis for subsequent processing steps. Subsequently, image enhancement is performed through a binarization process. Binarization entails converting the grayscale image into a black-and-white format. By setting a threshold, pixels with values above the threshold are rendered white, while those below are rendered black. With this step, the background areas with grayscale values below the threshold are set to white, while the ECGs are retained, as shown in Figure 3.

With these image processing methods, we can highlight key features, reduce noise, and eliminate unnecessary details, providing a clearer image foundation for subsequent analysis and diagnosis.

### 4.2. TETDiaNet

To our knowledge, clinicians make CAD diagnoses by observing the dynamic changes in the ECG under different states in the TET report. Based on this observation, we propose TETDiaNet to mimic this diagnostic process, as shown in Figure 4. The complete architecture of the proposed TETDiaNet is reported in Table 2.

In Figure 4, each sample contains three images with a 12-lead ECG. TETDiaNet first splits the image group into 12 parts according to the lead localization, and each part corresponds to a single lead. The split parts are assigned to different channels, i.e., 12×3=36 channels. The re-assigned features are fed into TETDiaNet, consisting of multiple TETDia blocks for feature extraction. The TETDia block is the core and basic unit of TETDiaNet, and it is composed of an intra-state contextual learning module and an inter-state contextual learning module. The TETDia block leverages the intra-state contextual learning module to investigate the temporal relationships within individual ECGs under a single state and the inter-state contextual learning module to learn the correlations between different states. Section 4.2.1 and Section 4.2.2 detail the structure of these modules within the TETDia block.

#### 4.2.1. Intra-State Contextual Learning Module

In the TETDia block, the intra-state contextual learning module explores the temporal relationships within individual ECGs, which is the foundation for the determination of the dynamic variations in the ECGs under different states.

Since different channels correspond to distinct ECGs, the intra-state contextual learning module leverages a convolution layer with dense groups to efficiently capture the complex temporal relationships within a single ECG. Specifically, this module divides the input channels into distinct groups, and each group contains only a single channel. For each group, the intra-state contextual learning module performs convolution separately within each group. This processing can be formulated as
(1)$${\mathrm{IR}}_{k,l,m}=\sum_{i,j}K_{i,j,m}\cdot F_{k+i-1,l+j-1,m},$$
where *K* is the convolutional kernel of size $D_{k}\times D_{k}\times M$ for the intra-state contextual learning module, *F* is the input feature map, and IR is the output of the module. The *m*-th filter in *K* is applied to the *m*-th channel in *F* to produce the *m*-th channel of the filtered output feature map IR.

Through dense grouping channels and independent convolutions, the intra-state contextual learning module can effectively learn complex patterns without introducing additional parameters. In addition, its computational complexity is proportional to the number of groups, making it efficient for ECG analysis within a single ECG.

#### 4.2.2. Inter-State Contextual Learning Module

The inter-state contextual learning module is the other important part of the TETDia block. It investigates the dynamic changes in the ECG under different states as the clinician does by learning the correlations between different states for each lead.

To achieve this, the inter-state contextual learning module leverages multiple 1×1 convolutions, as shown in Figure 4. Each 1×1 convolution aggregates the features of the same lead to learn the correlations of the lead between different states. Assuming that $\mathrm{IR}\in R^{H\times W\times C}$ is the input for this module, we can divide IR into 12 groups and suppose that the *g*-th group ${\mathrm{IR}}_{g}$ corresponds to the *g*-th lead. Then, 12 1×1 convolutions are used for inter-state contextual learning. For the *g*-th lead, its corresponding feature can be computed as
(2)$${\mathrm{IT}}_{k,l,g,m_{ou}}=\sum_{m_{in}}K_{1,1,g,m_{in},m_{ou}}^{IT}\cdot {\mathrm{IR}}_{k,l,g,m_{in}},$$
where $K^{IT}$ is the *g*-th 1×1 convolutional kernel of size $1\times 1\times M_{in}\times M_{ou}$ for the inter-state contextual learning module, ${\mathrm{IR}}_{g}$ is the input feature map (${\mathrm{IR}}_{g}$ represents the *g*-th group of IR), and IT is the output of the module (${\mathrm{IT}}_{g}$ denotes the *g*-th group of IT). $m_{in}$ and $m_{ou}$ indicate the input channel and the output channel of the *g*-th convolution.

The inter-state contextual learning module enables the TETDia block to adjust the feature representations across states. This is particularly beneficial in the context of 12-lead ECG images, where each channel corresponds to the specific ECG of an individual lead capturing distinct cardiac information. In other words, this module learns the correlations between different states for each lead, which helps the TETDia block to investigate the dynamic changes in the ECG under different states as a clinician does.

#### 4.2.3. Classifier

TETDiaNet learns discriminative features for CAD diagnosis by investigating the temporal relationships within individual ECGs under a single state and between different states. Based on the features learned by TETDiaNet, the proposed method employs a multilayer perceptron (MLP) as the classifier for CAD diagnosis.

Specifically, we flatten the features generated by TETDiaNet to obtain a one-dimensional vector. Then, this vector is fed into the classifier to produce a vector with a length of 2. A SoftMax function is utilized to normalize the vector, and each element of the normalized vector can be regarded as the probability that the patient has or does not have CAD. This processing can be defined as follows:(3)$${\hat{y}}_{i}=\mathrm{SoftMax}\left\{\mathrm{MLP}\left[\mathrm{Flatten}\left(F_{i}\right)\right]\right\},$$
where ${\hat{y}}_{i}$ is the diagnosis prediction of $F_{i}$, and *i* indicates the features and predictions. SoftMax denotes the SoftMax function. MLP(·) and Flatten(·) are the operations of the MLP classifier and flatten operation. The final prediction is determined by applying a threshold of 0.5. Predictions greater than or equal to 0.5 are classified as positive (CAD), while predictions below 0.5 are classified as negative (non-CAD).

### 4.3. Object Function

Since CAD diagnosis is a binary classification task, the cross-entropy loss is applied here as the object function. The proposed method optimizes the TETDiaNet and the classifier by minimizing this object function. The object function for the proposed method can be formulated as
(4)$$L=-\frac{1}{N}\sum_{i=1}^{N}\sum_{c=1}^{C}y_{i,c}\log\left({\hat{y}}_{i,c}\right),$$
where $y_{i,c}$ and ${\hat{y}}_{i,c}$ denote the ground-truth label and prediction of the *i*-th sample. c=1,2,…,C represents the CAD classes, and C=2. Based on Equation (Equation 4), we train an optimal TETDiaNet for CAD diagnosis.

## 5. Experiments

This section evaluates the diagnostic performance of the proposed method on the self-built TET4CAD dataset. Section 5.1 and Section 5.2 describe the TET4CAD dataset and evaluation metrics. Section 5.3 details the data augmentation and parameter settings. We compare the proposed method with the baseline methods mentioned in Section 5.4. Section 5.5 includes a visualization analysis using ROC curves and t-SNE to depict the contributions of the proposed modules to the performance improvement. In addition, we conduct a diagnostic analysis based on age and gender in Section 5.6.

### 5.1. Dataset

We conduct the subsequent experiments on the self-built TET4CAD dataset. This dataset was approved by Beijing Hospital (protocol number: 2024BJYYEC-KY043-01). In brief, 416 patients experiencing cardiac discomfort actively participated in data collection. Each participant underwent both the treadmill exercise test (TET) and coronary angiography (CAG) for CAD diagnosis, which contributed to the collection of TET data and the annotation of diagnosis labels. More details can be found in Section 5.1.

In the subsequent experiments, we randomly select 353 TET data from TET4CAD for the training set. This subset comprises 161 samples diagnosed with CAD through CAG, and the remaining 192 without CAD. Furthermore, the remaining 63 samples constitute the testing set. In the testing set, 31 samples are diagnosed with CAD, while 32 remain CAD-free. Table 3 provides more details of the training and testing sets.

### 5.2. Evaluation Metrics

To measure the diagnostic performance of the proposed method and the compared methods, a variety of evaluation metrics are applied here.

Some metrics leverage quantitative results to evaluate diagnosis methods, including accuracy, sensitivity, and specificity. Specifically, assuming that we have access to the predictions and true labels of *N* samples, we can be informed of the true negatives (*TN*) and true positives (*TP*). *TN* are instances where the model correctly predicts the absence of CAD when it is indeed absent in the actual data, while *TP* are instances where the model correctly predicts the presence of CAD when it is indeed present in the actual data. Meanwhile, we can obtain false negatives (*FN*), where the model incorrectly predicts the absence of CAD when it is actually present in the actual data, and false positives, where the model incorrectly predicts the presence of CAD when it is actually absent in the actual data. These three metrics can be defined as follows. Accuracy is a measure of the overall correctness of a diagnostic test and represents the ratio of correctly identified cases (both true positives and true negatives) to the total number of cases. Accuracy can be computed as
(5)$$\mathrm{accuracy}=\frac{TP+TN}{N}.$$

Regarding sensitivity, it is also known as the true positive rate or recall, and it gauges the ability of a diagnostic test to correctly identify individuals who truly have the disease. It is given by the formula
(6)$$\mathrm{sensitivity}=\frac{TP}{TP+FN}.$$
As for specificity, it measures the accuracy of a diagnostic test in correctly identifying individuals without the disease. Specificity can be computed via the following formula:(7)$$\mathrm{specificity}=\frac{TN}{TN+FP}.$$
Precision is the proportion of true positive predictions among all positive predictions made by a diagnostic or predictive model. Precision can be computed via the following formula:(8)$$\mathrm{precision}=\frac{TP}{TP+FP}.$$
Negative predictive value (NPV) is the proportion of true negative predictions among all negative predictions made by a diagnostic or predictive model, which can be computed via the following formula:(9)$$\mathrm{NPV}=\frac{TN}{TN+FN}.$$
All five of these are positive metrics, i.e., higher values of these metrics (accuracy, sensitivity, specificity, prevision, and NPV) correspond to better diagnosis performance.

In addition to these quantitative evaluation metrics, we also provide some visualization results for a qualitative comparison. The receiver operating characteristic (ROC) curve is a graphical representation used to assess the performance of a binary classification model across different thresholds. We apply it to illustrate the trade-off between the true positive rate (sensitivity) and the false positive rate (1-specificity) as the discrimination threshold for the classification of positive instances is varied. Moreover, t-distributed stochastic neighbor embedding (t-SNE) is employed to visualize high-dimensional data in a lower-dimensional space. Specifically, we output the features with a dimension of 256 from TETDiaNet and map them into a lower-dimensional space while preserving the pairwise similarities between points. This visualization of t-SNE effectively reveals the underlying structure and relationships within the data, which can show the contribution of the proposed method to the performance improvement.

### 5.3. Experimental Settings

In the subsequent experiments, we implement the proposed approach on a platform with RedHat 4.8.5, Xeon(R) E5-2620, a 1 TITAN X GPU, and PyTorch 1.8. The experimental settings are described in detail below.

#### 5.3.1. Baseline Methods

We take four representative machine learning approaches as the baseline methods for comparison, and they can be roughly divided into two categories. The first is a non-deep learning method, i.e., support vector machine (SVM) [41,42]. SVM is a supervised machine learning algorithm used for classification by finding a hyperplane that best separates the data into different classes or predicts a continuous target variable. The second category is deep learning methods, including AlexNet [43], GoogleNet [44], and ResNet18 [45]. AlexNet sparked this century’s trend of using deep learning to solve vision problems. It achieves recognition performance beyond that of the human eye by using a structure with five convolutional layers followed by max-pooling layers and three fully connected layers. GoogleNet presents an inception module, which concatenates filters of different sizes within the same layer. This architecture promotes the capture of both local and global features effectively. ResNet18 introduces residual connections, making it easier to train deeper models without suffering from vanishing or exploding gradients.

#### 5.3.2. Data Augmentation

To enrich the data diversity, we adapt the data augmentation operations employed in [12,14], including slight blurring and binarization with different thresholds generated by simple thresholding, Otsu thresholding, or adaptive thresholding.

#### 5.3.3. Parameters

For the baseline methods, the paper applies the hyperparameter settings from the published literature and retrains them under our evaluation protocols. As for the proposed method, Table 2 and Figure 4 show its structure. For a fair comparison, the related experiments employ the same optimizer and optimization strategy for all methods. The optimizer uses Adam with a learning rate of $10^{-5}$ for TETDiaNet and a learning rate of $10^{-4}$ for the classifier. In addition, the batch size is set to 30, and the maximum number of epochs is 2000.

### 5.4. Diagnostic Accuracy

To demonstrate the effectiveness of the proposed method, we conduct experiments on the self-built TET4CAD dataset. The baseline methods mentioned in Section 5.3 are employed as comparison methods, while the accuracy, sensitivity, and specificity detailed in Section 5.2 are the metrics applied to evaluate our method. We reproduce the compared methods by following the settings in Section 5.3 and apply the same preprocessing (Section 4.1) and data augmentation for a fair comparison. We perform five repetitions of the experiment for all methods and calculate their average accuracy, sensitivity, and specificity as the final experimental results. The results of the reproduced methods are assumed to be their best scores regarding diagnostic performance. Table 4 shows their performance, and we highlight the best results in bold.

According to the results in Table 4, the following conclusions can be reached.

**1. The newly built dataset, TET4CAD, has the potential to play a pivotal role in catalyzing advancements in CAD diagnosis**. The diagnostic capabilities of various methods on TET4CAD are systematically presented in Table 4. Notably, the experimental outcomes affirm the commendable diagnostic accuracy achieved by ResNet18 and our method, surpassing previous benchmarks [12,13]. This performance improvement can be attributed to the unique characteristics of the present dataset, which offers multiple ECGs recorded under diverse physiological states for the same patient. The inclusion of such varied ECG data allows machine learning methods to probe and leverage the dynamic variations in cardiac signals across different physiological states. This exploration of nuanced information contributes significantly to the enhanced performance observed in CAD diagnosis, underlining the potential of TET4CAD as a valuable resource to advance cardiovascular health research.

**2. The proposed TETDiaNet showcases exceptional performance when evaluated on the TET4CAD dataset**. The comprehensive comparison, as illustrated in Table 4, underscores the notable performance superiority of our algorithm over ResNet18 with the closest structure, namely ↑10.45%@accuracy in the relative value. The observed enhancement in performance can be ascribed to the algorithm’s adept exploration of the dynamic variation across different states in a single-lead ECG. Specifically, our algorithm explicitly delves into the temporal correlations between ECGs captured under the same physiological state and those recorded during different states, achieving this through the judicious combination of intra-state and inter-state contextual learning modules. The systematic exploration of these temporal correlations yields a more comprehensive understanding of the underlying patterns, contributing significantly to the algorithm’s heightened performance in contrast to ResNet18. This emphasis on the temporal interdependencies within the data is identified as a critical factor in achieving superior performance improvements, thereby reinforcing TETDiaNet’s efficacy in the context of CAD diagnosis on the TET4CAD dataset.

**3. The proposed TETDiaNet introduces a notable reduction in computational complexity when contrasted with ResNet18, as highlighted in Table 4**. The presented result unequivocally reveals a 72.74% decrease in the computational cost of the proposed method compared to Resnet18, the model with the closest structural affinity. This computational economy primarily emanates from the distinctive TETDia block integrated within TETDiaNet. In contrast to traditional convolution, the TETDia block divides ECG analysis into two specialized modules for CAD diagnosis, i.e., the intra-state contextual learning module and the inter-state contextual learning module. The former delves into the exploration of the temporal correlations within a single physiological state, while the latter explicitly investigates the temporal correlations between different physiological states. This innovative design not only contributes to an improvement in diagnostic performance but also concurrently reduces the computational complexity. The discerning separation of intra-state and inter-state analyses not only refines the diagnostic efficacy but also enhances the algorithmic speed, expanding the range of applicable scenarios for the TETDiaNet model.

### 5.5. Visualization Analysis

To provide a comprehensive qualitative evaluation of the proposed methodology, we employ ROC curves and t-SNE for visualization analysis. The ROC curves serve as a discerning metric, elucidating the superior performance of the proposed method through an analysis of the curve distributions. In contrast, t-SNE leverages an unsupervised clustering algorithm to visualize feature distributions in a two-dimensional space. This representation encapsulates both within-class compactness and cross-class discrimination, offering insights into the diagnostic efficacy of the proposed method. Figure 5 and Figure 6 illustrate the ROC curves and t-SNE, facilitating a qualitative comparison of the proposed methodologies and providing a nuanced understanding of their diagnostic capabilities.

**1. Regarding ROC curves**. As a common visualization analysis tool, the ROC curve illustrates the performance of an algorithm by showing the change in the true positive rate as the false positive rate increases. Notably, proximity to the upper-left corner of the ROC space correlates with enhanced diagnostic performance. The closer the curve is to the upper-left corner, the better the performance of the diagnosis method. In the context of our study, the ROC curve of the proposed TETDiaNet, as illustrated in Figure 5, shows close proximity to the upper-left corner. This positioning unequivocally signifies its superior performance over the comparative methods under consideration. This performance improvement comes from the TETDia block. It captures rich information for diagnosis by analyzing the temporal correlations within a single state and between different states. Figure 5 demonstrates the effectiveness of the proposed method and supports the above analysis.

Figure 5 also reports the area under curve (AUC) values of the compared methods and the proposed TETDiaNet in the lower-right area. Obviously, the proposed TETDiaNet has the highest AUC of 0.95, which demonstrates the superior diagnosis performance of the proposed method. Moreover, compared to the non-deep learning method (i.e., SVM), deep learning methods such as TETDiaNet, ResNet18, GoogleNet, and AlexNet exhibit higher AUC values. This demonstrates the great power of deep learning models for medical diagnosis.

In addition, Figure 5 indicates that the ROC curve of TETDiaNet is not always on the upper-left side of the other curves. Specifically, when the false positive rate (FPR) is greater than 0.76, the ROC curve of the proposed TETDiaNet lies on the lower side of the curves of ResNet18 and AlexNet. This illustrates that the proposed TETDiaNet is not the best choice when we require more accurate diagnosis performance under a high FPR (i.e., high accuracy under low misdiagnosis).

**2. Regarding t-SNE**. Figure 6 presents a visual depiction of the feature distribution of sample points, crucial for the qualitative analysis. By comparing it with the t-SNE of ResNet18, a noteworthy observation emerges, i.e., the proposed method showcases a more compact distribution of sample points, rendering it more amenable to segmentation by a hyperplane. This intraclass compactness and interclass separability in the distribution underscore the efficacy of TETDiaNet in distinguishing between patients with and without CAD based on their TET reports. This performance advance stems from the following two aspects. First, the TET4CAD dataset contributes significantly to this discriminative capability. It provides multiple ECGs of the same person and maintains the ECGs’ dynamic variation under diverse states in the TET report. Second, the proposed method not only captures temporal correlations within the same state but also explicitly explores temporal correlations across different states. This dual consideration enhances the algorithm’s ability to discern subtle variations in ECG patterns, affirming its effectiveness and efficiency in CAD diagnosis.

### 5.6. Diagnostic Analysis Based on Gender and Age

The TET report, being a composite reflection of an individual’s physiological response to exercise stress, is inherently influenced by various factors, including physical fitness, gender, and age. In an effort to deepen our understanding of the intricate relationship between machine diagnostics and these influencing factors, this section undertakes a focused diagnostic analysis based on gender and age. To systematically investigate these associations, volunteers are stratified into distinct groups based on their gender or age, facilitating separate diagnostic analyses for each subgroup. Table 5 shows the diagnostic results of the proposed method for different genders and different age groups.

**1. Regarding gender**. Table 5 indicates a discernible trend wherein the proposed diagnostic method exhibits superior performance for males, achieving accuracy of 91.27%, compared to 86.69% for females. This diagnostic gender bias is caused by the following two aspects. First, clinical experience suggests that gender does affect TET reports. The optimization of the proposed TETDiaNet relies solely on ECGs derived from TET reports. Therefore, TET reports affected by gender would lead to a discrepancy in the diagnostic accuracy of the proposed TETDiaNet for different genders. Second, TET4CAD provides more male samples than female samples. This scale discrepancy regarding the different genders would influence the optimization of the proposed method, leading to a diagnostic gender bias. In summary, the results in Table 5 robustly affirm the observed gender-specific diagnostic tendencies. In addition, this diagnostic gender bias illustrates that TETDiaNet does extract valuable and informative clues from the ECGs of TET reports, verifying the effectiveness of the proposed method for CAD diagnosis.

**2. Regarding Age**. In Table 5, we also observe a similar predictive bias for different ages. Specifically, individuals younger than 50 are more likely to be correctly diagnosed by TETDiaNet. On the one hand, this indicates the sensitivity of the proposed TETDiaNet to age-related factors. This susceptibility to age influences is attributed to the fact that our method solely relies on the ECGs in TET reports, which are known to be affected by age. Therefore, the diagnosis performance of the proposed method is driven by the age factor. On the other hand, the predictive bias for different ages also underscores the effectiveness of TETDiaNet in extracting informative dynamic variations in ECGs under different states from TET reports, improving the diagnosis accuracy.

### 5.7. Sensitivity Analysis

Section 5.4, Section 5.5 and Section 5.6 demonstrate the effectiveness of the proposed method via quantitative and qualitative experiments. However, there is an important hyperparameter in the proposed TETDiaNet, i.e., *g* in Equation (Equation 2). *g* defines the number of divided groups of the inter-state contextual learning module. We recommend g=12 considering that there are 12 leads in each ECG. To explore the impact of *g* on the diagnosis performance, this section presents a parameter sensitivity analysis by choosing *g* from 4,6,9,12,18 (these numbers can be divided by 36). Figure 7 plots the accuracy curve with respect to *g*.

In Figure 7, the accuracy curve initially exhibits a sharp increase followed by a gradual decrease with the increment in the parameter *g*, and we can obtain the highest accuracy of 87.53% at g=12. When *g* is smaller than 12, ECG features from different leads are assigned to the same group. In contrast, the ECG features from the same lead would occupy multiple groups when *g* is larger than 12. However, both cases would hinder the intra-state contextual learning module in the next layer to learn discriminative features for CAD diagnosis. Based on the above results and analysis, we recommend g=12 for the best diagnosis performance.

## 6. Conclusions

To prompt the development of machine CAD diagnosis, this paper proposes a treadmill exercise test (TET) dataset named TET4CAD and a new approach to CAD diagnosis. The TET4CAD dataset contains TET reports with annotations for 192 CAD patients and 224 non-CAD patients. To facilitate subsequent automatic CAD diagnosis, the dataset simplifies the TET report by retaining only the three electrocardiograms (ECGs) corresponding to three states, namely pretest, exercise, and recovery. Specialized clinicians label each TET report according to the corresponding coronary angiography (CAG) result. Regarding the proposed approach, it applies a pipeline of “preprocessing–feature extraction–classifier”. The preprocessing step converts the ECGs in PDF format to a group of images and employs a series of image processing methods to obtain clear ECGs for each patient. In the feature extraction step, our approach presents a new backbone network named TETDiaNet to investigate the temporal relationships within TET ECGs. Specifically, TETDiaNet designs a TETDia block as the basic unit. The TETDia block mimics the diagnostic process of clinicians to capture effective information for diagnosis. It comprises an intra-state contextual learning module and an inter-state contextual learning module, and they model the temporal relationships of ECGs within a single state and between states, respectively. Based on the relationships learned by the TETDia block, the proposed TETDiaNet can work like a clinician to analyze the TET report for CAD diagnosis. The experimental results on TET4CAD demonstrate the superior performance of the proposed approach and indicate that the temporal relationship within TET ECGs is a discriminative clue for CAD diagnosis. In the future, we will further explore the temporal relationships within the TET ECG for the better interpretability of the proposed method.

## Figures and Tables

**Figure 1 sensors-24-02705-f001:**
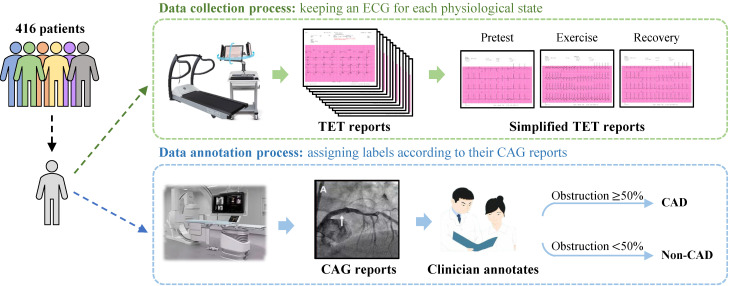
The illustration of the data collection and annotation of the TET4CAD dataset. The green box indicates the data collection of TET, while the blue box provides the details of annotation according to CAG. (“A” locates the basis for the clinician’s diagnosis of CAD based on this CAG report).

**Figure 2 sensors-24-02705-f002:**
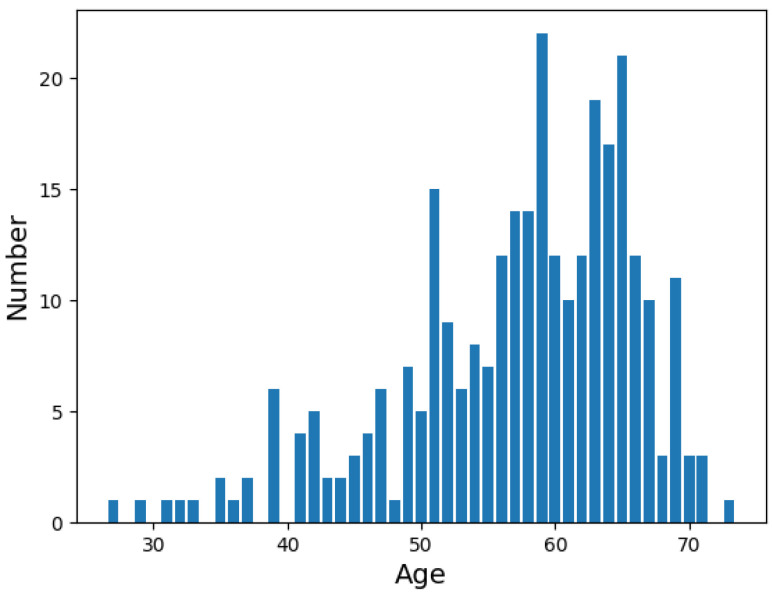
The illustration of the age distribution of the 416 patients in TET4CAD.

**Figure 3 sensors-24-02705-f003:**
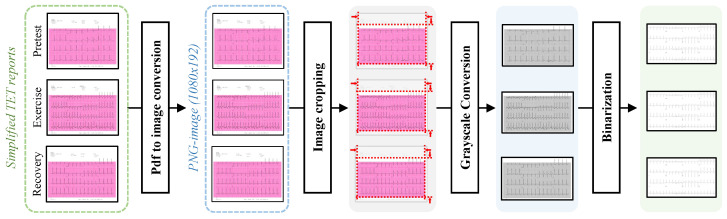
The pipeline of the preprocessing procedure. Initially, it transforms the TET report from its simplified PDF format into a set of three PNG images, optimizing them for further processing. Subsequently, the preprocessing module excises non-ECG regions through image cropping according to predefined coordinates. Finally, we convert the images to grayscale and apply binarization to augment the clarity and contrast of the images, thereby enhancing their suitability for subsequent analysis.

**Figure 4 sensors-24-02705-f004:**
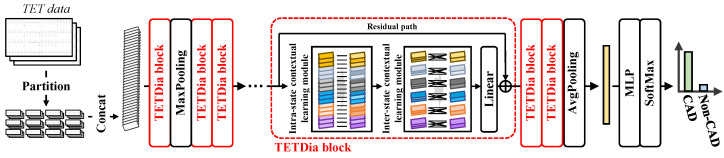
Structure of TETDiaNet. TETDiaNet leverages multiple TETDia blocks to mimic the process of clinicians diagnosing CAD based on the dynamic variations in the ECG under different states in the TET report. The TETDia block consists of an intra-state contextual learning module and an inter-state contextual learning module. The former explores the temporal relationships within individual ECGs, while the latter investigates the dynamic changes in the ECG under different states as the clinician does by learning the correlations between different states for each lead.

**Figure 5 sensors-24-02705-f005:**
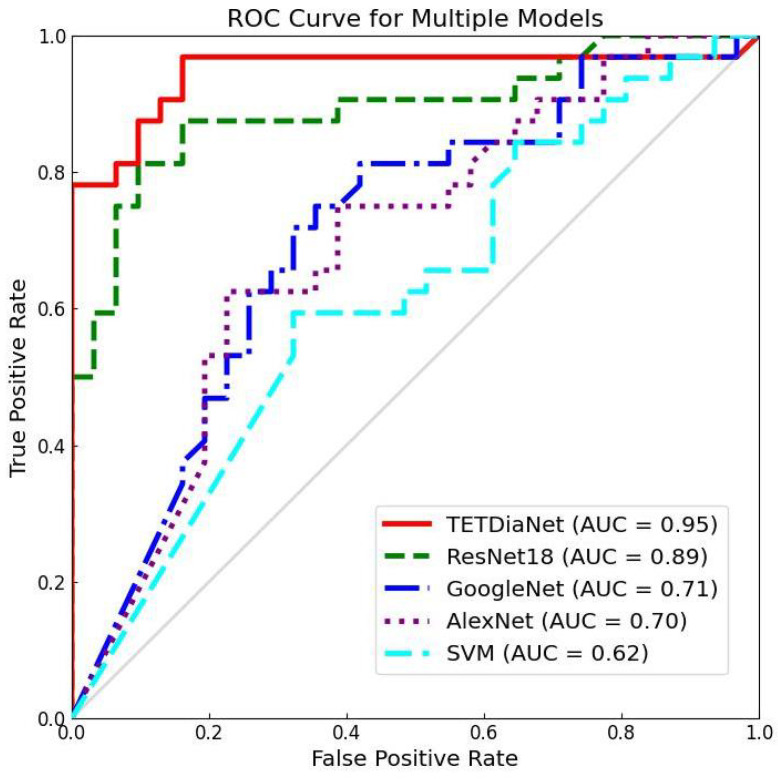
The comparison of the ROC curves of the compared method and the proposed TETDiaNet on the TET4CAD dataset.

**Figure 6 sensors-24-02705-f006:**
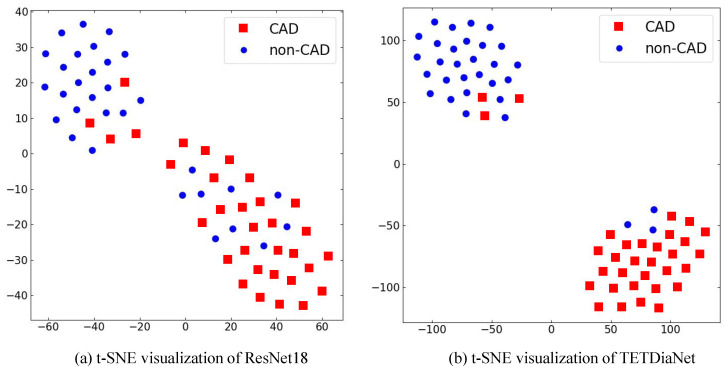
The comparison of the t_SNE visualizations between ResNet18 and the proposed TETDiaNet on the TET4CAD dataset.

**Figure 7 sensors-24-02705-f007:**
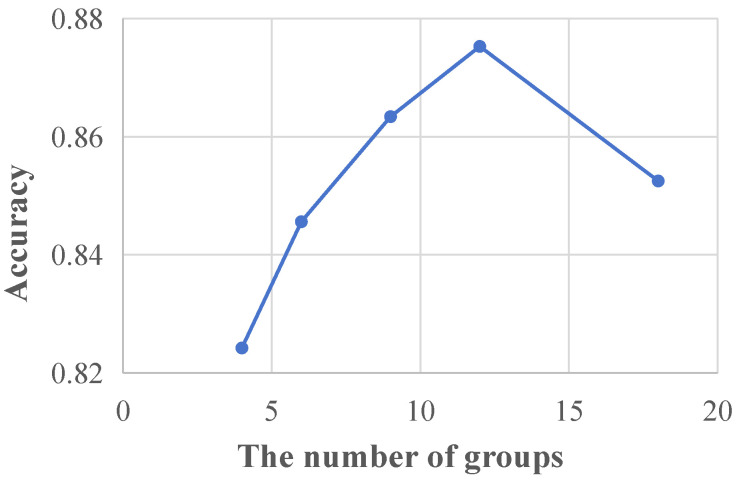
The sensitivity analysis of the number of groups (*g*) on the proposed TET4CAD dataset.

**Table 1 sensors-24-02705-t001:** Dataset characteristics and patient statistics of the TET4CAD dataset.

Characteristics	Sample Size (Total = 416)
Gender	–
Male	299
Female	117
Diagnostic label	–
Obstructive CAD	192
Non-obstructive CAD	224
Age	–
≤40	16
(40,50]	39
(50,60]	119
(60,70]	118
>70	4

**Table 2 sensors-24-02705-t002:** The architecture of TETDiaNet.

Type/Stride	Intra-State	Inter-State	Linear	Input Size
	$D_{k}\times D_{k}\times C$	(g)C×C	${\mathrm{in}}_{C}\times {\mathrm{out}}_{C}$	
TETDia block 1	7×7×1	(12)3×3	36×72	224×224×36
MaxPooling	-	-	-	56×56×72
TETDia block 2	3×3×1	(12)6×6	72×72	56×56×72
TETDia block 3	3×3×1	(12)6×6	72×72	56×56×72
TETDia block 4	3×3×1	(12)6×6	72×144	56×56×144
TETDia block 5	3×3×1	(12)12×12	144×144	28×28×144
TETDia block 6	3×3×1	(12)12×12	144×288	28×28×128
TETDia block 7	3×3×1	(12)24×24	288×288	14×14×288
TETDia block 8	3×3×1	(12)24×24	288×288	14×14×288
TETDia block 9	3×3×1	(12)24×24	288×288	14×14×288
TETDia block 10	3×3×1	(12)24×24	288×576	14×14×288
TETDia block 11	3×3×1	(12)48×48	576×576	7×7×576
TETDia block 12	3×3×1	(12)48×48	576×576	7×7×576
TETDia block 13	3×3×1	(12)48×48	576×576	7×7×576
AvgPool	-	-	-	7×7×576
Linear	-	-	576×128	1×1×576
Linear	-	-	128×2	1×1×128
SoftMax	-	-	-	1×1×2

**Table 3 sensors-24-02705-t003:** Training and testing sets for the TET4CAD Dataset.

Subset	Sample Size (Total = 416)
Training set	353
Obstructive CAD	161
Non-obstructive CAD	192
Testing set	63
Obstructive CAD	31
Non-obstructive CAD	32

**Table 4 sensors-24-02705-t004:** Quantitative comparison on the self-built TET4CAD dataset. (%) Note: (1) Five repetitive experiments were conducted and we report their averages in this table. (2) The **best** results are highlighted in bold.

Methods	Accuracy	Sensitivity	Specificity	Precision	NPV
SVM	57.35	75.64	39.27	55.45	61.71
AlexNet	62.64	81.52	42.63	58.69	69.76
GoogleNet	68.90	75.81	61.88	69.68	86.07
ResNet18	79.25	66.87	**90.01**	**87.00**	73.17
TETDiaNet (Ours)	**87.53**	**91.12**	84.89	85.78	**90.53**

**Table 5 sensors-24-02705-t005:** Diagnostic analysis of gender and age on the self-built TET4CAD dataset. (%) Note: (1). The results were obtained through five replicate experiments conducted. (2). The **best** results are highlighted in bold.

Group	Accuracy	Sensitivity	Specificity	Precision	NPV
**Gender**					
Male	**91.27**	**92.61**	79.84	82.12	**91.55**
Female	86.69	80.53	**92.14**	**91.11**	82.56
**Age**					
≤40	**100.00**	**100.00**	**100.00**	**100.00**	**100.00**
(40,50]	**100.00**	**100.00**	**100.00**	**100.00**	**100.00**
(50,60]	88.47	90.82	87.43	87.84	90.50
(60,70]	85.95	88.87	83.59	84.43	88.25
>70	80.00	83.33	75.00	76.92	81.82

## Data Availability

Dataset available on request from the corresponding author Deping Liu (lliudeping@263.net).

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
