# Peer review of "Temporal Relationship-Aware Treadmill Exercise Test Analysis Network for Coronary Artery Disease Diagnosis"

_sensors, 2024, doi:10.3390/s24092705_

Round 1

Reviewer 1 Report

Comments and Suggestions for Authors

The paper presents several strengths in its approach to computer-aided diagnosis (CAD) of cardiovascular diseases, particularly coronary artery disease (CAD). Firstly, the methodology introduced in the paper is innovative and offers a fresh perspective on CAD diagnosis by utilizing a Treadmill Exercise Test (TET) dataset. By leveraging this dataset, the paper proposes a comprehensive approach that includes preprocessing, feature extraction, and classification steps, providing a holistic framework for CAD diagnosis. Moreover, the paper demonstrates a meticulous approach to experimentation, with detailed descriptions of the dataset, evaluation metrics, experimental settings, and diagnostic analyses. This thoroughness enhances the credibility of the proposed method and facilitates a comprehensive understanding of its performance. Additionally, the paper employs visualization techniques such as ROC curves and t-SNE for insightful visualization analysis, allowing for a nuanced assessment of the method's diagnostic capabilities. Furthermore, the paper addresses the influence of gender and age on CAD diagnosis, offering valuable insights into the method's effectiveness across different demographic groups.

However, while the paper demonstrates several strengths in its approach to CAD diagnosis, there are also notable limitations that warrant attention. One limitation is the lack of detailed discussion on potential biases or confounding factors in the dataset, particularly regarding age and gender. Addressing these biases could enhance the generalizability of the proposed method across diverse demographic groups. Additionally, the paper could benefit from a more comprehensive comparison with existing state-of-the-art CAD diagnosis methods, including a broader range of baseline methods and performance metrics. Moreover, the experimental section could be expanded to include a sensitivity analysis of key parameters and hyperparameters, providing insights into the robustness of the proposed method. Furthermore, while the paper presents visualization analyses using ROC curves and t-SNE, additional interpretation and discussion of these results would enrich the paper's insights. Lastly, the paper could explore avenues for enhancing the interpretability of the proposed method, perhaps by incorporating explainable AI techniques to clarify the decision-making process. Addressing these limitations would strengthen the overall contribution of the paper to the field of CAD diagnosis.

Reviewer 2 Report

Comments and Suggestions for Authors The current manuscript develops the treadmill exercise test serving as a noninvasive method diagnostics of  coronary artery disease. Without doubts, the subject of the manuscript is extremely important and many people can potentially benefit from this material being published. Prior to its acceptance, however, a number of points need to be improved, please see the comments below.   In the sections and subsections where the image preparation, image analysis, and quantification of the statements are being performed i would propose---for a more general and less specialized audience (as the current referee)---to describe the steps also somewhat qualitatively so that each step of the procedure becomes understandable for an average reader (outside of the immediate area of investigation).   The structure in Fig. 4 seems very important, but the image itself is rather hard to read. Please add a clear and extensive caption of each of the steps presented there. Importantly, both in this diagram and in the entire text the authors should clarify to the reader the actual added value of their study. For instance, it should be clearly stated how the diagnostics is being conducted by standard tools nowadays (with all necessary details) and how it is proposed to be conducted now, with these new tools and learning techniques describe in the current manuscript.   Likewise, the quantifiers in Sec. 4.2.3 should be clarified to the reader in more details. Are these the standard ones and what are other options? For instance, for quantifying models of diffusion various machine-learning [https://iopscience.iop.org/article/10.1088/1367-2630/ab6065/meta], [https://www.nature.com/articles/s41467-021-26320-w] as well as some Bayesian algorithms have been employed in Ref. [https://doi.org/10.1039/C8CP04043E]. These three recent studies---as well as further investigations in the field of images of particle diffusion---should be cited in the revised version in order to provide a more clear picture to the reader regarding of what are some alternative statistical approaches for computer-based image-quantification analysis.   More details are to be presented also in Sec. 5.2, where the most important quantifiers are being evaluated and compared.   In Fig. 6 use not only different colors, but also different---and much larger---symbols for the data points, please.   Two last big sections have a strange sub-sectioning. Please keep the scheme of subsections there being the same as in the other parts of the manuscript. For a rather long and a richly structured paper i would strongly advise the authors to include the table of content prior to the introduction. I hope the rules of the journal will allow this: this will ease the life of the readers enormously.   Check typos in the text, please.
